# Toward Robust Image Manipulation Localization: A Novel Framework with VLMs and Weight-Aware Decoder

## Abstract

Image Manipulation Localization (IML) aims to identify and pinpoint regions within an image that have been forged or manipulated. Although some progress has been made in the task of IML, existing techniques still face several challenges. First, tampering techniques are diverse and complex, leaving various tampering artifacts in images. To effectively identify different types of tampered images, the model must extract comprehensive and highly discriminative tampering features. Second, some frameworks of IML use identical weights to fuse features from different scales during the decoding process, ignoring the varying sensitivity of different scales to the prediction results. To address these challenges, we propose a novel framework VLWA-Net, based on Vision-Language Models (VLMs). This framework leverages a VLMs-enhanced Artifact Extractor and a Multi-Domain Artifact Modulator to capture rich and discriminative tampering features, combining with traditional noise features as auxiliary cues. Next, we introduce a Weight-Aware Decoder (WAD) that comprehensively accounts for the sensitivity differences across scales and among feature points within the same scale. Additionally, the overall framework is trained using a Joint Information Supervision strategy, which enhances the model's ability to capture and perceive the details of tampered regions. The experimental results demonstrate that the proposed framework significantly improves accuracy on multiple mainstream test datasets and exhibits strong robustness and generalization capabilities.

## 1 Introduction

With the advancement of technology, an increasing number of tools and techniques allow for easily editing digital images. The proliferation of high-fidelity manipulation methods has made it increasingly difficult to distinguish between real and tampered visual content. This phenomenon not only undermines their authenticity but also causes adverse effects to fields such as news reporting and forensic evidence. Therefore, effectively localizing tampered regions in images is crucial for safeguarding the security of social information. However, existing frameworks of IML still face numerous challenges in addressing increasingly complex and diverse tampering techniques.

Firstly, tampering methods are highly diverse and sophisticated(Asghar et al., 2017; Alahmadi et al., 2013; Sadeghi et al., 2018; Chang et al., 2013), as illustrated in Fig. 1. Specifically, splicing typically leaves prominent discontinuities and lighting inconsistencies at image boundaries. Copy-move often generate repetitive textures, abnormal symmetry, and edge distortions. Removal usually results in unnatural transitions in color, lighting, and texture between the tampered and original regions, even causing subtle distortions. As can be seen, different tampering methods leave diverse artifacts on images. For a model to detect various types of tampered images, it must be capable of capturing comprehensive and highly discriminative tampering features. Compared to backbone based on CNNs(Liu et al., 2022b) and Transformers(Khan et al., 2022), VLMs exhibit superior universal feature extraction capabilitiesLi et al. (2025); Sun et al. (2025); Zhang et al. (2025a), enabling to capture richer tampering features. Moreover, the few-shot learning capability of VLMs enables them to adapt effectively to the task of IML. These factors highlight the immense potential of VLMs in enhancing the overall framework's localization accuracy, generalization, and robustness. However, only a few prior IML-related works(Zhang et al., 2024; 2025b; Su et al., 2024)

have employed VLMs, but they simply freeze the image encoder parameters for feature extraction. These approaches may introduce irrelevant features, weakening the discriminative power of the tampering features. Additionally, they lack direct and sufficient edge supervision for VLMs.

Secondly, some frameworks(Ma et al., 2024; Zhu et al., 2024) generally employ a uniform weighting strategy for multi-scale feature decoding, thereby neglecting the differences in sensitivity of features at various scales to the prediction outcomes. Features at different scales emphasize various aspects of tampering artifacts. A uniform weighting strategy often results in the dilution of some critical information, thereby limiting the model's localization accuracy.

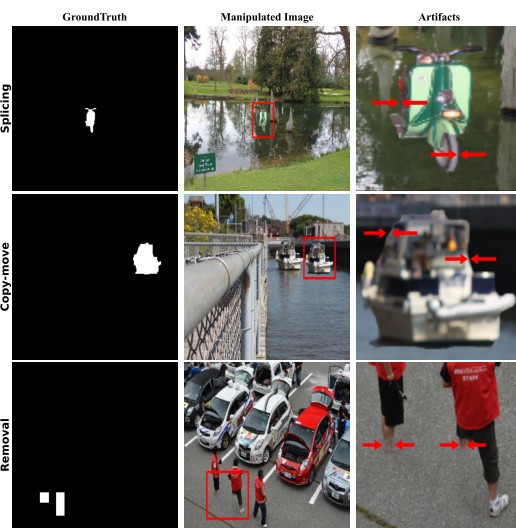

In addition, previous supervision methods(Guo et al., 2024; Zeng et al., 2024) mainly relied on segmentation region and edge information. Although these two types of supervisory information can reveal the structural differences in tampered regions, a multi-perspective supervision strategy can more comprehensively guide the model to learn tampering features and capture the multidimensional differences between authentic and tampered regions.

Figure 1: Examples of three types of tampering methods. The three columns, from left to right, represent groundtruth, tampered image, and the magnified artifact locations. For instance, in the upper right corner of the image, the splicing edge exhibits a noticeable unnatural texture transition.

All in all, state-of-the-art models of IML still commonly suffer from insufficient generalization and limited robustness in real-world applications. To address the above challenges, this paper proposes a novel framework named VLWA-Net, based on VLMs. This framework leverages the extensive pre-trained knowledge of VLMs to extract comprehensive and highly discriminative tampering features. A Multi-Domain Artifact Modulator is introduced to solve scale monotony and enhance feature representation. Meanwhile, commonly used noise tampering features are incorporated as auxiliary evidence. We further propose a Weight-Aware Decoder for predicting the segmentation mask. This decoder comprehensively accounts for the sensitivity differences among features at different scales and feature points within the same scale. Additionally, we employ a Joint Information Supervision strategy to train the framework, incorporating edge information, segmentation region information, and patch-level contrastive information. This multi-level supervision strategy encourages the model to capture both local and global tampering features more effectively.

In summary, the contributions of this paper are as follows:

- We propose a novel VLMs-based framework named VLWA-Net, which leverages a directly fine-tuned VLMs-enhanced Artifact Extractor to capture comprehensive and discriminative tampering features. To further improve the representation of tampering features, we introduce a Multi-Domain Artifact Modulator that enhances both the spatial and frequency components of features while increasing their scale diversity. The enhanced features are then combined with noise features for tampering region localization.

- To further enhance the accuracy of IML, we design a Weight-Aware Decoder that dynamically adjusts the decoding strategy based on the sensitivity of features at different scales and feature points within the same scale to the prediction results. This innovative component provides the model with greater flexibility in weight modulation.

- We propose a Joint Information Supervision strategy that deeply integrates segmentation region information, edge information, and patch-level contrastive information. This strategy enables the model to comprehensively learn the complex distribution differences between authentic and tampered pixels from multiple dimensions.

- Through extensive experiments on multiple publicly available datasets, it has been demonstrated that the proposed framework achieves significant progress in accuracy and robustness compared to existing state-of-the-art models. Meanwhile, our framework also has strong generalization ability.

## 2 RELATED WORKS

### 2.1 IMAGE MANIPULATION LOCALIZATION

Existing image manipulation localization frameworks can be broadly categorized into traditional methods(Kumar Singh et al., 2022; Ferrara et al., 2012; Verdoliva et al., 2014; Yuan, 2011) and deep learning-based methods(Wang et al., 2022b; Gao et al., 2022; Liu et al., 2024a; Dong et al., 2022; Liu et al., 2022a; Wang et al., 2022a). Traditional methods primarily rely on manually designed feature extractors. For instance, (Ferrara et al., 2012) develop a technique based on Color Filter Array (CFA) that calculates pixel interpolation errors between original and tampered areas. These handcrafted feature extractors capture tampering features that are singular and fixed, rendering them ineffective in detecting tampered images with subtle, diverse, and complex artifacts.

With the development of deep learning, Convolutional Neural Networks (CNNs) and Transformers have been introduced into the task of IML. CNN-based models, due to their strong feature extraction capabilities, can capture various tampering features. MVSS-Net(Dong et al., 2022) uses a dual-stream CNN network separately extract RGB features and noise features for jointly locating tampered regions, while PSCC-Net (Liu et al., 2022a) utilizes a CNN encoder to extract multi-scale features in a top-down manner and subsequently generates the tampering mask in a bottom-up, coarse-to-fine fashion. Vision Transformer (ViT)(Dosovitskiy et al., 2020) is a milestone work that introduced the transformer architecture to the computer vision. ViT, with its global self-attention mechanism and excellent semantic feature extraction capability, has offered a new perspective for the IML. Objectformer(Wang et al., 2022a) utilizes a ViT-based framework which leverages object prototypes to model object-level consistency. TruFor(Guillaro et al., 2023) feeds both an RGB image and learnable noise-sensitive fingerprints into a transformer-based fusion architecture to extract high-level and low-level features. Recently, some researchers have integrated the advantages of CNNs and Transformers to propose hybrid architectures. Mesorch(Zhu et al., 2025) deeply discuss how to technically characterize the artifacts exist at the mesoscopic level and propose a hybrid model combining CNNs and Transformers to efficiently construct the mesoscopic tampering features. However, these methods still struggle with insufficient generalization and robustness.

### 2.2 VISION-LANGUAGE MODELS

In recent years, VLMs have demonstrated groundbreaking advancements in computer vision by integrating multimodal semantic understanding capabilities. The large-scale parameters and extensive training data endow them with powerful general feature extraction capabilities and few-shot transfer learning ability, enabling effective modeling and understanding of complex visual information. CLIP(Radford et al., 2021) and ALIGN(Jia et al., 2021) utilize contrastive learning to embed images and text into a shared feature space, enabling the model to understand semantic relationships between visual and textual data, thereby achieving few-shot image classification and cross-modal retrieval. In the semantic segmentation domain, SAM(Kirillov et al., 2023) implement cross-scenario interactive segmentation with real-time responsiveness by integrating prompt engineering. CLIP and SAM, as foundational models, have inspired numerous derivative models for various downstream tasks in computer vision. (Liu et al., 2024b) fine-tune the MLP layers of the CLIP image encoder using Mixture-of-Experts, thereby achieving precise AIGC image detection. Med-SA(Wu et al., 2025) is a medical image segmentation model which embeds adaptive blocks at specific locations in the SAM image encoder and is trained with multimodal prompts. Therefore, the powerful feature extraction and few-shot transfer learning capabilities of VLMs can provide a new insight for overcoming performance bottlenecks in IML and enhancing both generalization and robustness.

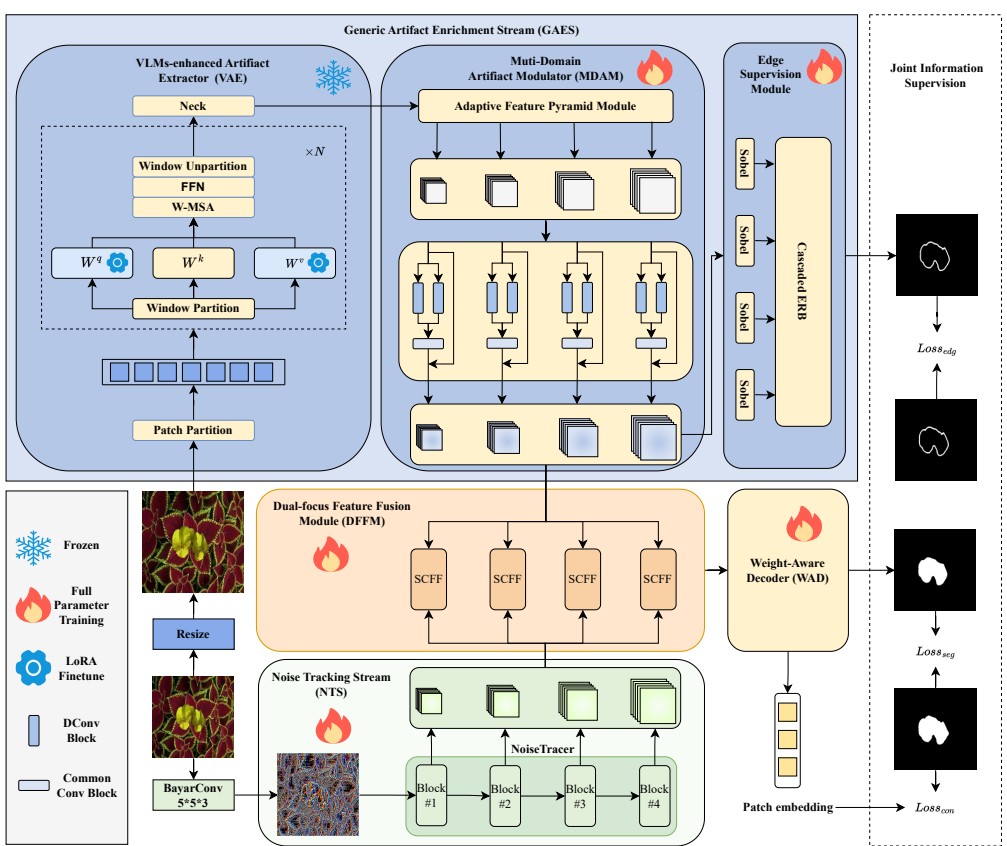

Figure 2: Pipeline design of VLWA-Net. The high-resolution image is fed into the GAES, where comprehensive and highly discriminative tampering features are extracted and enhanced. Local noise inconsistencies inherent in tampered images are captured by NTS, while a deep fusion of tampering features is performed by DFFM. These features are then sent to WAD to generate the final segmentation mask. The framework employs a Joint Information Supervision strategy to compute the loss.

## 3 METHODOLOGY

### 3.1 OVERALL FRAMEWORK

In this section, we will primarily introduce the details of the VLWA-Net, as shown in Fig. 2. The framework is composed of four main components: the Generic Artifact Enrichment Stream (GAES), Noise Tracking Stream (NTS), Dual-focus Feature Fusion Module (DFFM), and Weight-Aware Decoder (WAD). The framework starts with an RGB image, which is converted to a high-resolution image $I_H \in \mathbb{R}^{1024 \times 1024 \times C}$ and a noise distribution maps $I_N \in \mathbb{R}^{H \times W \times 3}$ using Bayar convolution kernel(Bayar & Stamm, 2018). The $I_H$ is processed by the GAES, where comprehensive and discriminative tampering features are extracted through a VLMs-enhanced Artifact Extractor (VAE). Subsequently, we employ a Multi-Domain Artifact Modulator (MDAM) to increase feature scale diversity and strengthen both spatial and frequency components. Meanwhile, the $I_N$ is fed into the NTS, which uses ConvNeXt(Liu et al., 2022b) as the NoiseTracer network to extract noise inconsistencies between real and tampered regions. Then, multi-scale features derived from the dual-flow architecture are deeply fused by the DFFM at both spatial and channel levels. Finally, WAD decodes the precise segmentation mask from the fused features and outputs patch embeddings for contrastive learning.

## 3.2 GENERIC ARTIFACT ENRICHMENT STREAM

The outstanding performance of VLMs(Kirillov et al., 2023) in semantic segmentation highlights their immense potential in the task of IML. Therefore, we propose a GAES including a VLMs-enhanced Artifact Extractor (VAE) that leverages its extensive pre-trained knowledge and excellent feature extraction capabilities to capture comprehensive and subtle tampering features. The VAE, as shown in Fig. 2, uses the MAE(He et al., 2022) to reconstruct the original image patches, which allows it to focus more on the structural relationships between pixels and object-level features. The encoding process is represented by the following equation:

$$F_R = \mathcal{N}\Big(\mathcal{B}\big(\mathcal{P}(I_H) \oplus P_e\big)\Big) \tag{1}$$

Where $\mathcal{N}$ is denoted as the output convolutional block, $\mathcal{B}$ refers to the ViT block using windowed self-attention, $\mathcal{P}$ represents the patch partitioning operation, and $P_e$ is the position embeddings. In this process, high-resolution RGB images $I_H$ are used as input. For image preprocessing, a direct resize is applied to increase the resolution, which helps preserve more detailed information and finer tampering traces. The VAE can then extract rich and discriminative tampering features $F_R \in \mathbb{R}^{\frac{H}{16} \times \frac{W}{16} \times C_{dim}}$ from these images. To retain the VAE's ability to capture general features while enhancing its learning of tampering features and reducing interference from irrelevant features, we employ the LoRA(Hu et al., 2022) method for fine-tuning.

To enrich the scale diversity of the features extracted by the VAE and enhance both spatial and frequency components to capture more tampering traces, we propose a Multi-Domain Artifact Modulator (MDAM). In the MDAM, the single-scale features output by the VAE are first sent to an Adaptive Feature Pyramid Module for multi-scale transformation. This module is composed of one upsampling branch and two downsampling branches. The upsampling operation is performed using a transposed convolution, followed by a convolutional block to eliminate checkerboard artifacts and align the feature dimensions with the NTS output for subsequent feature fusion. The downsampling branches utilize max pooling to extract the most responsive feature values, followed by convolutional blocks for feature reconstruction and alignment. Next, we enhance each scale's features in both the spatial and frequency domains. Specifically, for frequency domain enhancement, a Discrete Cosine Transform (DCT) is used to extract the

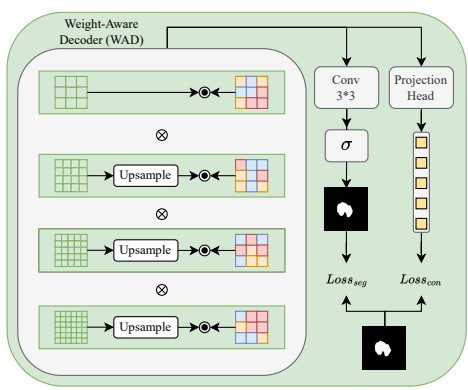

Figure 3: The design of WAD. WAD adaptively aggregates tampering features from four scales and then outputs a segmentation mask and patch embeddings for segmentation region supervision and patch-level contrastive supervision.

high-frequency components of the input features, followed by a convolution with a dilation rate of 4 to reconstruct the frequency-domain features. For spatial domain enhancement, we apply the same dilated convolution to deeply capture tampering features in the spatial domain. Finally, the spatial and frequency domain features are concatenated, fused using a convolutional block, and added to the input features to obtain the final output. Additionally, it is widely acknowledged that a significant amount of tampering artifacts exist along the boundary between tampered and authentic regions. To ensure that VAE and MDAM focus on these tampering edges, an Edge Supervision Module (ESM) is applied with sobel operators and a cascaded ERB(Dong et al., 2022). Ultimately, GAES outputs four scales of features $\{f_{r_1}, f_{r_2}, f_{r_3}, f_{r_4}\}$ along with a predicted tampering edge mask $\hat{P}_{edge} \in \mathbb{R}^{H \times W \times 1}$:

$$\{f_{r_1}, f_{r_2}, f_{r_3}, f_{r_4}\} = \text{MDAM}(\mathbf{F}_R),$$
$$f_{r_i} \in \mathbb{R}^{\frac{H}{2^{(i+1)}} \times \frac{W}{2^{(i+1)}} \times C_i} \tag{2}$$
$$\hat{P}_{edge} = \sigma(ESM(f_{r_1}, f_{r_2}, f_{r_3}, f_{r_4})))$$

Where the $\sigma$ denotes the sigmoid function, $C_i$ refers to the number of channels in the $i$-th ($i = 1, 2, 3, 4$) scale feature.

### 3.3 NOISE TRACKING STREAM

Most tampering techniques alter the noise traces of the original region or introduce new noise, resulting in a distinct difference in the noise distribution between the authentic and tampered regions. Inspired by this, we design the NTS to assist in IML. We opt for the trainable Bayar convolution kernel as our noise extractor to capture better noise distribution maps. Given that noise inconsistency is a non-semantic, localized feature, we choose the ConvNeXt as the NoiseTracer network to extract the multi-scale noise difference features $\{f_{n_1}, f_{n_2}, f_{n_3}, f_{n_4}\}$:

$$\{f_{n_1}, f_{n_2}, f_{n_3}, f_{n_4}\} = \text{NoiseTracer}\,(\mathbf{I}_N)\,,$$
$$f_{n_i} \in \mathbb{R}^{\frac{H}{2^{(i+1)}} \times \frac{W}{2^{(i+1)}} \times C_i} \tag{3}$$

To more comprehensively and deeply fuse the same-scale feature maps from GAES and NTS, we employ DFFM (the details are in appendix) for feature fusion.

### 3.4 WEIGHT-AWARE DECODER

Some existing models resort to simple operations such as scale alignment, element-wise addition, or concatenation during multi-scale feature decoding. These methods fail to fully account for the differences in sensitivity between features at various scales, as well as among feature points within the same scale, for prediction results. Therefore, we propose a Weight-Aware Decoder (WAD) with its details in the Fig. 3.

The decoder introduces four groups of learnable 3D weight parameters that can adaptively capture the sensitivity differences between features across different scales, as well as among pixels within the same scale. Through end-to-end training, these weight parameters can dynamically adjust the contributions of features from each scale, thereby better accommodating diverse tampering patterns and enhancing the model's ability to fit the training data. Specifically, a corresponding 3D weight matrix is computed for each scale's features. By performing element-wise multiplication between the fused features and the corresponding weight parameters, adaptive feature weighting is achieved. Finally, the weighted multi-scale features are concatenated and passed through a 3×3 convolutional block for further feature integration and tampered region prediction. The computational process is presented as follows:

$$\hat{P}_{seg} = \sigma(Conv(Concat(W_i \times f_i^{'}, i = 1, 2, 3, 4))) \tag{4}$$

Here, $\hat{P}_{seg} \in \mathbb{R}^{H \times W \times 1}$ denotes the predicted segmentation mask. $W_i$ is the weight matrix corresponding to the $i$-th scale feature. The projection head is used to map the concatenated features to a higher-dimensional feature space for contrastive learning.

### 3.5 JOINT INFORMATION SUPERVISION

We introduce a Joint Information Supervision strategy that enforces supervision on three distinct levels. Dice Loss(Milletari et al., 2016) is employed for both edge supervision $L_{edg}$ and segmentation region supervision $L_{seg}$ to alleviate the severe class imbalance between positive and negative samples. Additionally, traditional binary contrastive learning(He et al., 2020) is applied for patch-level contrastive supervision $L_{con}$. To reduce memory consumption, we divide the feature map into $n \times n$ feature patches and perform inter-patch contrastive learning. Specifically, we compute the patch embedding by averaging the embeddings of all pixels within the patch. The patch's label is determined based on the majority label of the pixels within the corresponding region in the groundtruth:

$$Dist_{q_i} = \frac{\exp\left(q_i \cdot b^+/\tau\right)}{\exp\left(q_i \cdot b^+/\tau\right) + \sum_{b^- \in N_i} \exp\left(q_i \cdot b^-\right)}$$
$$L_i = \frac{1}{|P_i|} \sum_{b^+ \in P_i} -\log Dist_{q_i} \tag{5}$$
$$L_{con} = \frac{1}{n^2} \sum_{i \in n^2} L_i$$

Here, $P_i$ represents the set of all patch embeddings $b^+$, whose embeddings share the same label as $q_i$. Similarly, $N_i$ denotes the set of all negative patch embeddings $b^-$ that have a different label from $q_i$. All embeddings in the loss function are $L_2$-normalized. For a single image, the contrastive loss is obtained by averaging the loss across all embedded image patches. Finally, the total loss function of the model is defined as follow. In our experiments, we set $\alpha = 0.01$ and $\beta = 0.19$.

$$L_{total} = \alpha * L_{con} + \beta * L_{seg} + (1 - \alpha - \beta) * L_{edg} \tag{6}$$

## 4 EXPERIMENT

### 4.1 EXPERIMENTAL SETUP

**Training Data and Implementation Details** Our model is trained on the standardized Protocol-CAT dataset (Kwon et al., 2021), which comprises CASIAv2 (Dong et al., 2013), IMD2020(Novozamsky et al., 2020), FantasticReality, and TampCOCO, totaling 825,997 images. These images encompass various tampering techniques, including splicing, copy-move and removal. Each training image is resized to both $512 \times 512$ and $1024 \times 1024$ resolutions for input. We train our model for 80 epochs on two NVIDIA 4090 GPUs. The learning rate adopts a cosine decay strategy, initialized at 1e-4 and progressively decreasing to a minimum of 5e-7. We utilize the AdamW optimizer with a weight decay of 0.05 to prevent overfitting.

**Test Dataset and Evaluation Metric** We evaluate our model using public benchmarks, covering seven widely used datasets: CASIAv1 (Dong et al., 2013), Coverage (Wen et al., 2016), NIST16 (Guan et al., 2019), Columbia (Hsu & Chang, 2006), COCOGlide(Guillaro et al., 2023), AutoSplice(Jia et al., 2023) and DSO (De Carvalho et al., 2013). These datasets consist of tampered images with varying resolutions and diverse tampering techniques. Additionally, we adopt the pixel-level F1 score and AUC score as evaluation metrics to quantitatively measure the performance of our model in IML. All our experiments are conducted using the standard threshold of 0.5.

Table 1: The performance comparison results are based on pixel-level F1 scores. The best-performing values are highlighted in bold, while the second-best are underlined.

| Method | Pixel-level F1 score | | | | | | | |
|---|---|---|---|---|---|---|---|---|
| | CASIA1 | Columbia | NIST16 | COVER | DSO | COCOGlide | AutoSplice | Average |
| MVSS-Net(Dong et al., 2022) | 0.583 | 0.723 | 0.320 | 0.470 | 0.355 | 0.428 | 0.388 | 0.466 |
| PSCC-Net(Liu et al., 2022a) | 0.578 | 0.822 | 0.416 | 0.341 | 0.345 | 0.458 | 0.455 | 0.487 |
| CAT-Net(Kwon et al., 2022) | 0.778 | 0.923 | 0.450 | 0.485 | 0.334 | 0.409 | 0.450 | 0.547 |
| TruFor(Guillaro et al., 2023) | 0.700 | 0.903 | 0.426 | 0.379 | 0.335 | 0.504 | 0.504 | 0.535 |
| SAM(Kirillov et al., 2023) | 0.627 | 0.817 | 0.509 | 0.401 | 0.519 | 0.574 | 0.381 | 0.546 |
| IML-ViT(Ma et al., 2024) | 0.751 | 0.927 | 0.140 | 0.546 | 0.453 | 0.369 | 0.343 | 0.504 |
| APSC-Net(Qu et al., 2024) | 0.798 | 0.941 | 0.500 | 0.402 | 0.617 | 0.190 | 0.506 | 0.565 |
| SparseViT(Su et al., 2025) | **0.827** | 0.959 | 0.384 | 0.513 | 0.239 | 0.386 | 0.387 | 0.527 |
| Mesorch(Zhu et al., 2025) | 0.826 | 0.905 | 0.412 | 0.526 | 0.446 | 0.397 | 0.357 | 0.552 |
| VLWA-Net(ours) | 0.811 | **0.961** | **0.584** | **0.619** | **0.751** | **0.588** | **0.514** | **0.690** |

### 4.2 PERFORMANCE COMPARISON WITH STATE-OF-THE-ART

For the state-of-the-art comparison, we select CATNet(Kwon et al., 2022), MVSSNet, PSCCNet, TruFor, IML-ViT, SAM, APSC-Net(Qu et al., 2024), SparseViT(Su et al., 2025) and Mesorch as our comparison baselines. To ensure a fair comparison, all these models are retrained on the Protocol-CAT dataset. We then evaluate their performance alongside our proposed VLWA-Net using the same metrics on public benchmark datasets. The experimental results are summarized in Table 1. From the table, it is evident that VLWA-Net outperforms existing state-of-the-art models in IML across all five benchmark datasets. Remarkably, our method demonstrates superior performance on high-resolution datasets such as NIST16 and DSO. The Fig. 4 presents a high-quality comparison of prediction results from our model and other state-of-the-art models. Evidently, our framework predicts tampered region boundaries with greater accuracy, achieving both lower false alarm rate and higher precision. The fact demonstrates that our framework effectively captures comprehensive

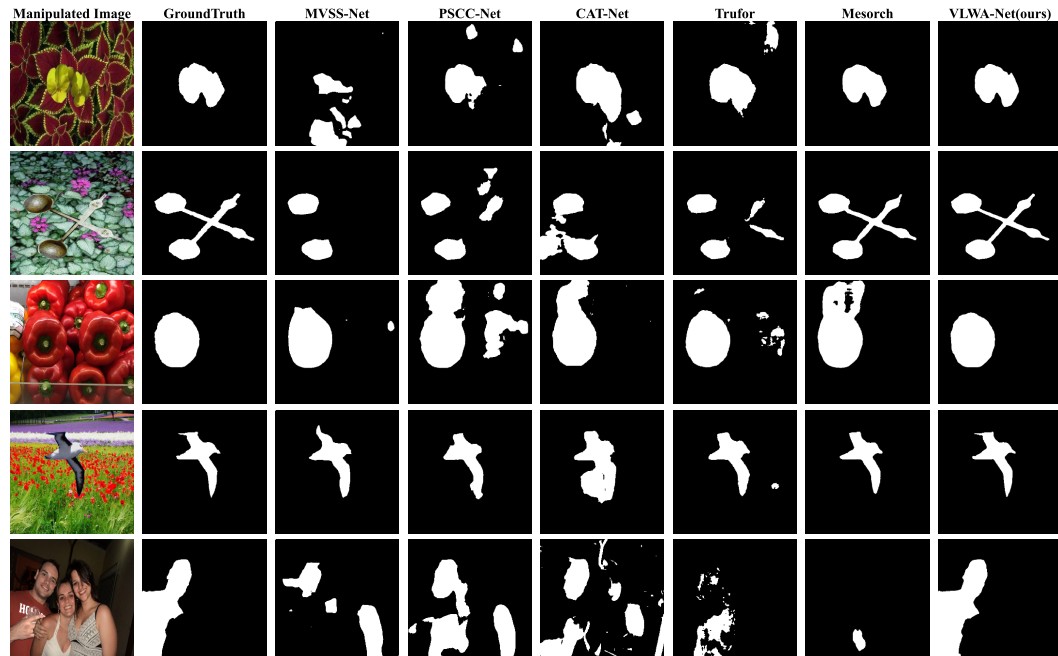

Figure 4: Manipulation localization results on images originating from multiple datasets. The left-most two columns are the manipulated image and ground truth mask, followed by the prediction results of different models.

and high-quality tampering artifacts and accurately localizes tampered regions by leveraging multi-scale weighted features.

### 4.3 ROBUSTNESS EVALUATION

To evaluate the model's performance under various attack conditions, we conduct robustness tests on the CASIAv1, Columbia, Coverage, and NIST16 datasets, with the results presented in Fig. 5. We apply degradation techniques such as Gaussian noise with different standard deviations, Gaussian blur with varying kernel sizes, and JPEG compression with different quality factors to the tampered images. It can be observed that regardless of the attack type, our framework outperforms other state-of-the-art methods on the NIST16 and Columbia datasets. For CASIAv1 and Coverage, our model exceeds the performance of other models under JPEG compression and Gaussian blur and is only slightly inferior to CAT-Net under Gaussian noise attacks. Overall, our model demonstrates strong robustness.

### 4.4 ABLATION STUDIES

Additionally, we conduct eight sets of experiments to comprehensively evaluate the impact of the proposed components on the overall framework performance. In Table 2, we report the average pixel-level F1 scores for each experiment across five publicly available benchmark datasets.

**Influence of VAE and MDAM** We first replace the VAE and MDAM in GAES with alternative backbones based on CNNs and Transformers, followed by performance evaluations. The results, as shown in settings 1, 2, 3 and 4 in Table 2, indicate that our VAE and MDAM achieve an improvement of approximately 83.5% over ConvNeXt, 26.9% over SegFormer and 13.9% over Dinov3. This fact demonstrates that our VAE and MDAM are capable of extracting comprehensive and highly discriminative tampering features.

**Influence of WAD** We maintain the other modules unchanged while replacing the decoder with a convolutional decoder(Zhu et al., 2024) in setting 6, an additive decoder in setting 5, a MLP

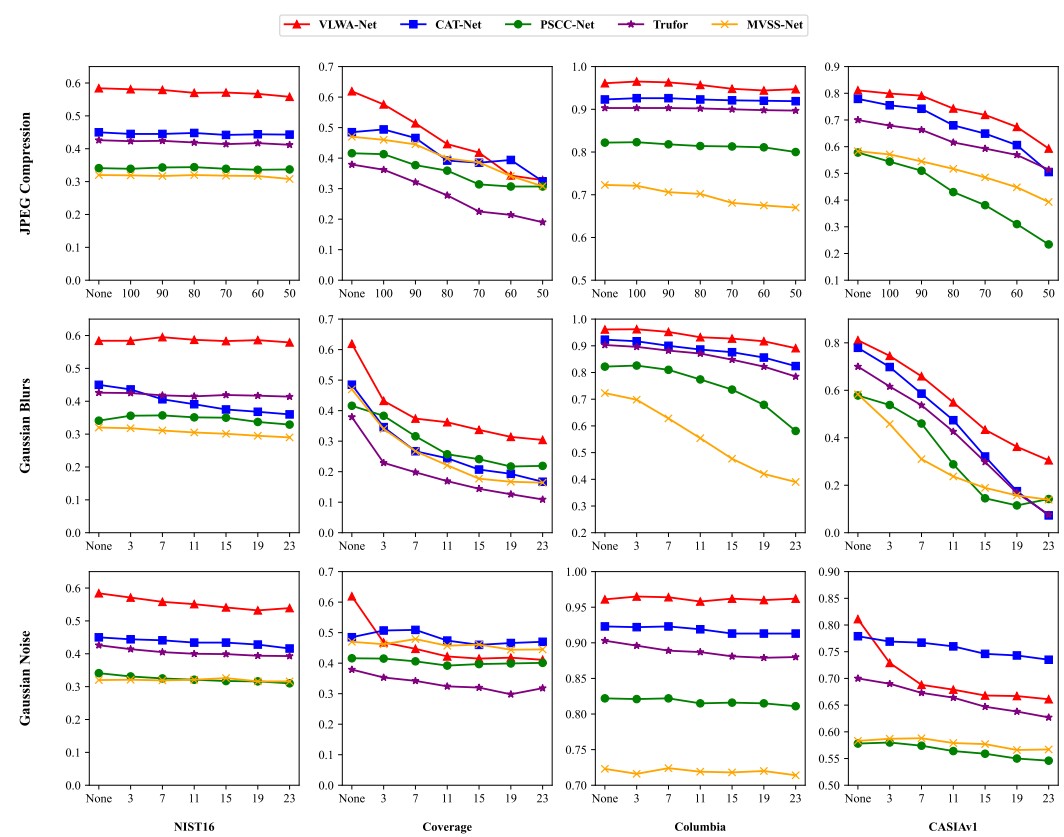

Figure 5: Robustness test results. We employ three types of attacks: JPEG compression, Gaussian noise, and Gaussian blur, using CASIA1.0, Coverage, Columbia, and NIST16 as the test datasets. The x-axis represents the attack intensity, while the y-axis denotes the pixel-level F1 score on the corresponding test datasets.

Table 2: Ablation Study Results. We remove the GAES, WAD, and Joint Information Supervision (JIS) components to evaluate their impacts on overall model performance.

| | components | | | Avg.F1 |
|---|---|---|---|---|
| | GAES | Decoder | JIS | |
| 1 | VAE+MDAM | WAD | edg+seg+con | **0.745** |
| 2 | ConvNeXt(Liu et al., 2022b) | WAD | edg+seg+con | 0.406 |
| 3 | SegFormer(Xie et al., 2021) | WAD | edg+seg+con | 0.587 |
| 4 | Dinov3(Siméoni et al., 2025) | WAD | edg+seg+con | 0.654 |
| 5 | VAE+MDAM | ADD | edg+seg+con | 0.722 |
| 6 | VAE+MDAM | Conv | edg+seg+con | 0.683 |
| 7 | VAE+MDAM | MLP | edg+seg+con | 0.701 |
| 8 | VAE+MDAM | SE Block | edg+seg+con | 0.710 |
| 9 | VAE+MDAM | WAD | edg+seg | 0.709 |

decoder(Ma et al., 2024) in setting 7 or a SE Block(Hu et al., 2018) in setting 8. The data in Table 2 indicates that our WAD improves the average F1 score by 9.1%, 3.2%, 6.3% and 4.7% respectively. The fact further demonstrates that WAD enhances the overall framework's performance by comprehensively accounting for the sensitivity differences among features at different scales and feature points within the same scale.

**Influence of Joint Information Supervision**   To demonstrate that our Joint Information Supervision strategy compels the model to learn more diverse and accurate distribution differences between authentic and tampered pixels, we replace our supervision strategy with commonly used edge supervision and segmentation region supervision adopted by other models in setting 9. Compared to setting 9, our Joint Information Supervision strategy improves the overall model performance by 5.1%.

## 5    CONCLUSION

In this paper, we explore the immense potential of VLMs for IML. We introduce the VLWA-Net, which leverages the VAE to capture comprehensive and highly discriminative tampering features. Moreover, the MDAM is used to increase feature scale diversity and strengthen both spatial and frequency components. Furthermore, drawing insights from the limitations of previous methods in decoding multi-scale features, we propose a Weight-Aware Decoder that accounts for sensitivity differences across different scales and among feature points within the same scale. Additionally, by introducing a Joint Information Supervision strategy, we effectively enhance the model's ability to capture subtle tampering features. Comparison experiments on standard benchmark datasets demonstrate that our approach not only surpasses current baseline models in pixel-level F1 score and AUC score but also exhibits excellent robustness and generalization capabilities.

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

# A  APPENDIX

## A.1  TECHNICAL NOVELTY

Indeed, only a few prior IML-related works have employed VLMs, but they simply freeze the image encoder parameters or rely on an adaptive block for feature extraction. These approaches may introduce irrelevant features, weakening the discriminative power of the features. In contrast, we design the GAES to capture comprehensive and discriminative tampering features, which is based on VLMs. The GAES differs from the following three aspects in its utilization of VLMs:

- We integrate the VLMs into the VAE to enhance the model's capability in extracting tampering features. By fine-tuning the VAE with LoRA, we dynamically adjust the feature space to focus on manipulation traces while suppressing irrelevant features, thereby significantly improving feature discriminability.

- To further improve the representation of tampering features output by VAE, we introduce the MDAM that enhances both the spatial and frequency components of features while increasing their scale diversity.

- We impose direct edge supervision on GAES to enhance focus on tampered boundaries.

Additionally, we designed DFFM and WAD to integrate the tampering features extracted by VLMs and noise features, consolidating all evidence for precise tampering localization.

Moreover, we innovatively integrate edge supervision, segmentation region supervision, and block-level contrastive information supervision with weighted fusion. In summary, VLMs serve as the foundation of our framework, and we design other modules around the VLMs.

## A.2  DETAILS OF THE LORA FINE-TUNING OF VAE

The $q$ and $v$ matrices in each ViT block of the VAE are fine-tuned using LoRA. Equation 2 represents the computation process of the $q$ and $v$ matrices before fine-tuning, while equation 3 represents the computation process after fine-tuning.

$$Q = W_q x \quad V = W_v x \tag{7}$$

$$Q = W_q x + B_q A_q x \quad V = W_v x + B_v A_v x \tag{8}$$

Here, $x \in \mathbb{R}^d$ denotes the tampering features, $W \in \mathbb{R}^{d \times d}$ denotes the original pre-trained weights, and $A$ and $B$ are a pair of low-rank matrices with rank $r(r \ll d)$.

## A.3  DETAILS OF THE DFFM

To more comprehensively and deeply fuse the same-scale feature maps from GAES and NTS, we employ DFFM for feature fusion. DFFM is comprised of four Spatial-Channel Feature Fusion blocks (SCFF). Each SCFF incorporates two parallel attention mechanisms: spatial attention and channel attention. This dual attention mechanism computes the similarity of feature vectors at any two positions and any two channels to generate attention maps $M_i^{PA}$ and $M_i^{CA}$, which is subsequently used to weight the feature map. Thus, it captures long-range dependencies between pixels, thereby emphasizing critical spatial and channel features. The computation process of each SCFF is

Table 3: The performance of VLWA-Net trained with 20% training data. Using pixel-level F1 score as evaluation metric.

| Method | Pixel-level F1 score | | | | | | | |
|--------|--------|----------|--------|-------|-----|-----------|------------|---------|
|        | CASIA1 | Columbia | NIST16 | COVER | DSO | COCOGlide | AutoSplice | Average |
| VLWA-Net | 0.759 | 0.919 | 0.545 | 0.558 | 0.659 | 0.654 | 0.549 | 0.663 |

as follows:

$$
\begin{aligned}
f_{input_i} &= Concat(f_{r_i}, f_{n_i}) \\
M_i^{CA} &= softmax(f_{input_i}^T \otimes f_{input_i}) \\
M_i^{PA} &= softmax(f_{input_i} \otimes f_{input_i}^T) \\
f_i^{CA} &= f_{input_i} \otimes M_i^{CA} \oplus f_{input_i} \\
f_i^{PA} &= M_i^{PA} \otimes f_{input_i} \oplus f_{input_i} \\
f_i' &= f_i^{CA} + f_i^{PA}
\end{aligned}
\tag{9}
$$

Here, $\otimes$ denotes matrix multiplication, and $\oplus$ represents element-wise addition. $f_i' \in \mathbb{R}^{\frac{H}{2^{(i+1)}} \times \frac{W}{2^{(i+1)}} \times 1}$ corresponds to the fused feature at the $i$-th scale ($i = 1, 2, 3, 4$).

## A.4 DETAILS OF THE EXPERIMENTAL SETUP

**Training Data and Implementation Details**  Our model is trained on the standardized Protocol-CAT dataset, which comprises CASIAv2, IMD2020, FantasticReality, and TampCOCO, totaling 825,997 images. These images encompass various tampering techniques, including splicing, copy-move and removal. Each training image is resized to both $512 \times 512$ and $1024 \times 1024$ resolutions for input. We train our model for 80 epochs on two NVIDIA 4090 GPUs. During each epoch, 14,070 images are randomly selected from the training dataset with a global batch size of 2. The learning rate adopts a cosine decay strategy, initialized at 1e-4 and progressively decreasing to a minimum of 5e-7, accompanied by a 2-epoch warmup period for stable parameter initialization. We utilize the AdamW optimizer with a weight decay of 0.05 to prevent overfitting. To improve training stability, gradient accumulation is implemented with 16 steps, equivalently scaling the effective batch size while maintaining generalization capability across diverse data distributions.

**Test Dataset and Evaluation Metric**  We evaluate our model using public benchmarks, covering seven widely used datasets: CASIAv1, Coverage, NIST16, Columbia, COCOGlide, AutoSplice and DSO. These datasets consist of tampered images with varying resolutions and diverse tampering techniques. Additionally, we adopt the pixel-level F1 score and AUC score as evaluation metrics to quantitatively measure the performance of our model in IML.

## A.5 FEW-SHOT LEARNING CAPABILITY ANALYSIS OF VLWA-NET

Using 20% of the original training set as a new training dataset, we retrained our model and evaluated it on seven public test sets. The experimental results are in Table 3. The results demonstrate that our model maintains superior overall performance compared to other models, indicating that we have fully leveraged the few-shot transfer capability of VLMs and achieved strong sample efficiency or dependency on data scale.

## A.6 AUC RESULTS REPORT

We report the pixel-level AUC scores of the most advanced IML models, as shown in Table 4. From the table, it is evident that VLWA-Net outperforms existing state-of-the-art models in IML across all seven benchmark datasets. Remarkably, our method demonstrates superior performance on high-resolution datasets such as NIST16 and DSO. The fact demonstrates that our framework effectively captures comprehensive and high-quality tampering artifacts and accurately localizes tampered regions by leveraging multi-scale weighted features.

Table 4: The performance comparison results are based on pixel-level AUC scores. The best-performing values are highlighted in bold, while the second-best are underlined.

| Method | Pixel-level AUC score | | | | | | | |
|---|---|---|---|---|---|---|---|---|
| | CASIA1 | Columbia | NIST16 | COVER | DSO | COCOGlide | AutoSplice | Average |
| MVSS-Net | 0.904 | 0.911 | 0.777 | 0.868 | 0.772 | 0.819 | 0.755 | 0.829 |
| PSCC-Net | 0.918 | 0.919 | 0.810 | 0.872 | 0.811 | 0.848 | 0.879 | 0.865 |
| CAT-Net | 0.965 | 0.962 | 0.867 | 0.907 | 0.836 | 0.849 | 0.862 | 0.893 |
| TruFor | 0.951 | 0.936 | 0.863 | 0.887 | 0.854 | 0.888 | 0.908 | 0.898 |
| SAM | 0.945 | 0.973 | 0.876 | 0.886 | 0.944 | 0.874 | 0.849 | 0.906 |
| IML-ViT | 0.961 | 0.941 | 0.812 | 0.921 | 0.838 | 0.835 | 0.854 | 0.881 |
| APSC-Net | 0.916 | 0.942 | 0.775 | 0.731 | 0.744 | 0.578 | 0.714 | 0.771 |
| SparseViT | 0.963 | 0.974 | 0.851 | 0.919 | 0.855 | 0.863 | 0.881 | 0.901 |
| Mesorch | **0.979** | 0.924 | 0.891 | 0.917 | 0.912 | 0.894 | **0.926** | 0.920 |
| VLWA-Net(ours) | 0.967 | **0.979** | **0.904** | **0.922** | **0.965** | 0.898 | 0.889 | **0.932** |

Table 5: The detection performance comparison results are based on image-level F1 scores. The best-performing values are highlighted in bold.

| Method | Image-level F1 score | | | | | |
|---|---|---|---|---|---|---|
| | CASIA1 | Columbia | NIST16 | COVER | DSO | Average |
| MVSS-Net | 0.798 | 0.636 | **1.000** | 0.667 | **1.000** | 0.800 |
| PSCC-Net | 0.581 | 0.709 | 0.971 | 0.641 | **1.000** | 0.780 |
| TruFor | 0.336 | 0.066 | 0.826 | 0.578 | 0.930 | 0.547 |
| VLWA-Net(ours) | **0.819** | **0.714** | 0.994 | **0.675** | **1.000** | **0.841** |

## A.7 DETECTION PERFORMANCE COMPARISON WITH STATE-OF-THE-ART

Many real-world applications require a fast, image-level decision before performing expensive pixel-level analysis. So, we add a detection head to our original model and retrain it using the MVSS-protocol. We select MVSS-Net, PSCC-Net, and TruFor as baselines (since only these models possess image-level classification capability) and retrain them on the same dataset. The models are evaluated on CASIA1.0, Columbia, Coverage, NIST16, and DSO as test sets. The first three contain both authentic and tampered images, while the last two contain only tampered images. The image-level F1-scores and Accuracy are reported in the Table 5 and 6. The results demonstrate that our model significantly outperforms these baseline models on both metrics.

## A.8 PERFORMANCE COMPARISON ON GAN-BASED TAMPERING METHODS

We evaluate our model on the ForgeryADE dataset, which contains images tampered with by four mainstream GAN models. It is worth noting that our training set does not contain GAN-based tampered images. The test results are presented in Table 7. As can be seen, our method achieves SOTA performance. Future work will focus on further enhancing the model's capability in localizing GAN-based manipulations.

## A.9 SENSITIVITY OF PERFORMANCE TO THE CHOICE AND INTENSITY OF SIMULATED PERTURBATIONS DURING TRAINING

During the training process, we apply various augmentation techniques to the images including scale transformation, random copy-move, random inpainting, spatial transformations, color/brightness enhancements, quality degradation, and blurring. Each technique is applied to the images with a certain probability. We retrain our method after reducing the types and intensity of the simulated perturbations. The results are shown in Table 8. The reduced perturbations lead to a 3.6% performance drop. Therefore, we conclude that VLWA-Net's performance advantage is not highly sensitive to this factor. This also demonstrates a certain degree of training stability in our model.

## A.10 COMPARATIVE ANALYSIS OF DIFFERENT VLMS

We evaluate the performance of the VLWA-Net architecture using different VLMs with various sizes. The results are presented in Table 9. We replace the VAE with a ViT-L-sized CLIP and use

Table 6: The detection performance comparison results are based on image-level Acc scores. The best-performing values are highlighted in bold.

| Method | Image-level Acc score | | | | | |
|---|---|---|---|---|---|---|
| | CASIA1 | Columbia | NIST16 | COVER | DSO | Average |
| MVSS-Net | 0.535 | 0.466 | **1.000** | 0.500 | **1.000** | 0.700 |
| PSCC-Net | 0.646 | 0.618 | 0.943 | 0.485 | **1.000** | 0.738 |
| TruFor | 0.224 | 0.181 | 0.704 | 0.425 | 0.870 | 0.481 |
| VLWA-Net(ours) | **0.797** | **0.644** | 0.988 | **0.590** | 1.000 | **0.804** |

Table 7: The performance comparison results are based on pixel-level F1 scores. The best-performing values are highlighted in bold.

| Method | Pixel-level F1 score | | | | |
|---|---|---|---|---|---|
| | Crfill(Zeng et al., 2021) | Ctsdg(Guo et al., 2021) | Deepfillv2(Yu et al., 2019) | LaMa(Suvorov et al., 2021) | Avg.F1 |
| MVSS-Net | 0.202 | 0.223 | 0.245 | **0.231** | 0.225 |
| PSCC-Net | 0.210 | 0.224 | 0.222 | 0.225 | 0.220 |
| CAT-Net | 0.205 | 0.220 | 0.222 | 0.217 | 0.216 |
| IML-ViT | 0.203 | 0.222 | 0.218 | 0.214 | 0.214 |
| Mesorch | 0.212 | 0.216 | 0.241 | 0.224 | 0.223 |
| TruFor | 0.202 | 0.221 | 0.219 | 0.217 | 0.215 |
| VLWA-Net(ours) | **0.215** | **0.226** | **0.252** | 0.222 | **0.229** |

two input resolutions: 224×224 and 336×336. The model achieves pixel-level average F1 scores of 0.534 and 0.551 under the two settings, slightly below the latest SOTA model. We hypothesize this stems from CLIP's primary design for image classification tasks, lacking specialized training for fine-grained semantic segmentation. When we replace the VAE with a ViT-B-sized SAM, the model achieves an average F1-score of 0.586, surpassing all other SOTA models. Compared to the ViT-L-sized SAM setting, this setup features fewer parameters and faster inference speed. This fact demonstrates that our framework fully unleashes the potential of VLMs for IML tasks.

## A.11  THE IMPACT OF JOINT INFORMATION SUPERVISION ON MODEL ROBUSTNESS

We compare the robustness of the two configurations under various attacks, including Gaussian noise (GN), Gaussian blur (GB), JPEG compression (JC), scaling perturbations, and their combinations. The evaluation is conducted on the CASIA1.0 dataset using the average pixel-level F1-score as the metric. The results are presented in Table 10. The results demonstrate that incorporating Joint Information Supervision(JIS) leads to a marked improvement in robustness, particularly against Gaussian blur attacks.

## A.12  VISUAL ANALYSIS OF WAD

We conduct a visual analysis of the four sets of weight parameters in WAD and generated corresponding heatmaps in Fig. 6. The heatmaps reveal that the magnitude (e.g., 0.1, 0.001) and value ranges of the weights differ across each set of parameters, and even within the same set, the parameter values include both positive and negative values. This sufficiently demonstrates that WAD comprehensively considers the sensitivity differences of multi-scale features to the final results, emphasizing discriminative features while suppressing irrelevant ones.

## A.13  FLOPS AND PARAMETERS

The number of parameters and FLOPs for all measurements was calculated based on a batch size of 1. As shown in Table 11, our model has a comparable computational burden to VLMs-based models while demonstrating higher accuracy.

Table 8: The results are based on pixel-level F1 scores. The best-performing values are highlighted in bold.

| Method | Pixel-level F1 score | | | | | |
|---|---|---|---|---|---|---|
| | CASIA1 | Columbia | NIST16 | COVER | DSO | Average |
| VLWA-Net w/ reduced perturbations | 0.783 | 0.956 | 0.565 | 0.570 | 0.716 | 0.718 |
| VLWA-Net(ours) | **0.811** | **0.961** | **0.584** | **0.619** | **0.751** | **0.745** |

Table 9: The performance of the same module uses different VLMs. "Avg.F1" represents the mean value of the standard F1 score across seven datasets.

| VLMs | Size | Avg.F1 |
|---|---|---|
| SAM | ViT-L | 0.690 |
| | ViT-B | 0.586 |
| CLIP | ViT-L/14 | 0.534 |
| | ViT-L/14@336px | 0.551 |

Table 10: The performance comparison results are based on average pixel-level F1 scores using CASIA1.0 as test dataset. The best-performing values are highlighted in bold.

| Method | Average pixel-level F1 score | | | | | | | |
|---|---|---|---|---|---|---|---|---|
| | None | GN | GB | JC | GN+JC | GB+GN | GB+JC | Scale |
| VLWA-Net w/o JIS | 0.785 | 0.600 | 0.408 | 0.661 | 0.652 | 0.252 | 0.168 | 0.400 |
| VLWA-Net(ours) | **0.811** | **0.700** | **0.552** | **0.733** | **0.723** | **0.397** | **0.321** | **0.414** |

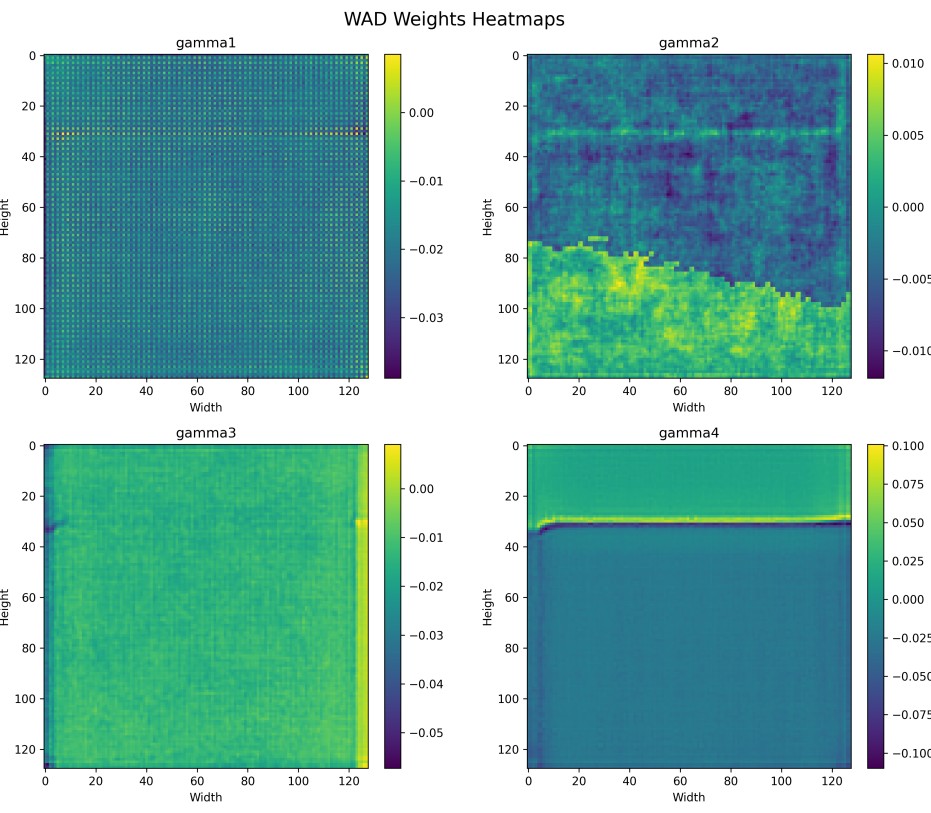

Figure 6: Heatmap visualization of four groups 3D Weight Parameters of WAD

Table 11: Comparison of parameters and computational efficiency (Flops) across different models. VLWA-Net* represents VLWA-Net using SAM-B.

| Model | Parameters (M) | FLOPs (G) |
|---|---|---|
| SAM | 309 | 1499 |
| IMDPrompt(Zhang et al., 2025b) | 347.6 | 1533 |
| VLWA-Net | 482 | 1667 |
| VLWA-Net* | 263 | 659 |

