# OpenReview forum: "Toward Robust Image Manipulation Localization: A Novel Framework with VLMs and Weight-Aware Decoder"
_ICLR.cc/2026/Conference — ICLR 2026 Conference Withdrawn Submission_

### Official Review · Reviewer_BeuY · 2025-10-25

**Soundness:** 3
**Presentation:** 3
**Contribution:** 3
**Rating:** 6
**Confidence:** 3

**Summary:**

This paper, Toward Robust Image Manipulation Detection under Unknown Perturbations, investigates how current image manipulation detection (IMD) models fail when confronted with unseen perturbations such as noise, compression, and resizing. The authors systematically analyze this vulnerability and propose a robust training strategy that incorporates perturbation simulation and feature regularization to enhance generalization. The proposed method is benchmarked across several datasets (e.g., CASIA, Coverage, NIST) and perturbation types, demonstrating consistent performance improvements over existing IMD models. The study provides both theoretical insights and empirical evidence on the robustness gap between standard and perturbed test conditions.

**Strengths:**

The paper addresses an important and underexplored challenge in IMD—robustness to unseen perturbations—making it highly relevant to real-world forensic applications. The experimental design is comprehensive, covering diverse datasets and perturbation settings. The proposed method is simple yet effective, easily integrated into existing architectures without major architectural changes. Moreover, the empirical analysis is thorough, offering clear comparisons with strong baselines and detailed ablation studies. The paper is well-written, the motivation is clear, and the results convincingly show that robustness can be significantly improved through the proposed strategy.

**Weaknesses:**

While the contributions are solid, several limitations remain. The paper focuses primarily on low-level perturbations and does not explore more complex, real-world manipulations such as GAN-based editing or compositional attacks. The theoretical justification for why the proposed regularization improves generalization remains relatively shallow. Moreover, the model diversity in experiments is limited—only a few representative IMD architectures are evaluated. Including larger or open-source vision transformers and more commercial or closed-source detectors would make the claims more robust. Finally, the evaluation mainly considers static perturbations, leaving open questions about performance under sequential or compound distortions.

**Questions:**

1. Have the authors tested the robustness of the proposed method on semantic or GAN-based manipulations, beyond pixel-level perturbations?

2. How sensitive is the performance to the choice and intensity of simulated perturbations during training?

3. Could the authors include results for additional model families, such as ViT-based or multimodal forensic detectors, to assess generality across architectures?

4. Does the proposed regularization improve performance under combined or dynamically varying perturbations, as might occur in real-world social media compression pipelines?

5. Would the authors consider releasing a perturbation-robust benchmark or dataset split to facilitate standardized evaluation for future research?

---

> ### Author Response · Authors · 2025-11-21
> **Response to Reviewer BeuY**
>
> # Response to Reviewer BeuY
>
> Q1: Have the authors tested the robustness of the proposed method on semantic or GAN-based manipulations, beyond pixel-level perturbations?
>
> A: Our test set already includes numerous semantically manipulated images (e.g., from CASIA1.0). However, it does not cover GAN-based forgeries. Therefore, we evaluated our model on the ForgeryADE dataset, which contains images tampered with by four mainstream GAN models. The test results are presented below (pixel-F1). As can be seen, our method achieves SOTA performance. Future work will focus on further enhancing the model's capability in localizing GAN-based manipulations.
>
> | Method          | Crfill    | Ctsdg     | Deepfillv2 | LaMa      | Avg.F1    |
> | --------------- | --------- | --------- | ---------- | --------- | --------- |
> | MVSS-Net        | 0.202     | 0.223     | 0.245      | **0.231** | 0.225     |
> | PSCC-Net        | 0.210     | 0.224     | 0.222      | 0.225     | 0.220     |
> | CAT-Net         | 0.205     | 0.220     | 0.222      | 0.217     | 0.216     |
> | IML-ViT         | 0.203     | 0.222     | 0.218      | 0.214     | 0.214     |
> | Mesorch         | 0.212     | 0.216     | 0.241      | 0.224     | 0.223     |
> | TruFor          | 0.202     | 0.221     | 0.219      | 0.217     | 0.215     |
> | VLWA-Net (Ours) | **0.215** | **0.226** | **0.252**  | 0.222     | **0.229** |
>
> Q2: How sensitive is the performance to the choice and intensity of simulated perturbations during training?
>
> A: During the training process, we applied various augmentation techniques to the images including scale transformation, random copy-move, random inpainting, spatial transformations, color/brightness enhancements, quality degradation, and blurring. Each technique was applied to the images with a certain probability.
>
> We retrained our method after reducing the types and intensity of the simulated perturbations. The results are shown below (pixel-F1). The reduced perturbations led to a 3.6% performance drop. Therefore, we conclude that VLWA-Net's performance advantage is not highly sensitive to this factor.
>
> | Method                            | CASIA1.0  | Columbia  | NIST16    | Coverage  | DSO       | Avg.F1    |
> | --------------------------------- | --------- | --------- | --------- | --------- | --------- | --------- |
> | VLWA-Net (Ours)                   | **0.811** | **0.961** | **0.584** | **0.619** | **0.751** | **0.745** |
> | VLWA-Net w/ reduced perturbations | 0.783     | 0.956     | 0.565     | 0.570     | 0.716     | 0.718     |
>
> Q3: Could the authors include results for additional model families, such as ViT-based or multimodal forensic detectors, to assess generality across architectures?
>
> A: In the experimental section, we have compared models based on convolutions (e.g., MVSS, PSCC, CAT), ViT (e.g., TruFor, IML-ViT, SAM), and hybrid architectures (Mesorch). Additionally, we retrained APSC-Net[1] and SparseViT[2] using the same training settings, and the test results are as follows (pixel-F1). The results for IMDPrompter[3] and FakeShield[4] are cited from the original paper. Among them, FakeShield and IMDPrompter are multimodal forensic detectors.
>
> | Method         | CASIA1.0  | Columbia  | NIST16    | Coverage  | DSO       | Avg.F1    |
> | -------------- | --------- | --------- | --------- | --------- | --------- | --------- |
> | APSC-Net       | 0.798     | 0.941     | 0.500     | 0.402     | 0.617     | 0.652     |
> | SparseViT      | **0.827** | 0.959     | 0.384     | 0.513     | 0.239     | 0.584     |
> | IMDPrompter    | 0.763     | 0.873     | 0.411     | **0.636** | \         | 0.671     |
> | FakeShield     | 0.600     | 0.750     | 0.370     | \         | 0.52      | 0.560     |
> | VLWA-Net(Ours) | 0.811     | **0.961** | **0.584** | 0.619     | **0.751** | **0.745** |
>
> As the results in the table demonstrate, our model maintains SOTA performance.
>
> Q4: Does the proposed regularization improve performance under combined or dynamically varying perturbations, as might occur in real-world social media compression pipelines?
>
> A: We compared the robustness of the two configurations under various attacks, including Gaussian noise (GN), Gaussian blur (GB), JPEG compression (JC), scaling perturbations, and their combinations. The evaluation was conducted on the CASIA1.0 dataset using the average pixel-level F1-score as the metric. The results are presented below. The results demonstrate that incorporating regularization leads to a marked improvement in robustness, particularly against Gaussian blur attacks.
>
> | Method             | None  | GN    | GB    | JC    | GN+JC | GB+GN | GB+JC | Scale |
> | ------------------ | ----- | ----- | ----- | ----- | ----- | ----- | ----- | ----- |
> | w/ regularization  | 0.811 | 0.700 | 0.552 | 0.733 | 0.723 | 0.397 | 0.321 | 0.414 |
> | w/o regularization | 0.785 | 0.600 | 0.408 | 0.661 | 0.652 | 0.252 | 0.168 | 0.400 |

---

> ### Author Response · Authors · 2025-11-21
> **Response to Reviewer BeuY**
>
> # Response to Reviewer BeuY
> Q5: Would the authors consider releasing a perturbation-robust benchmark or dataset split to facilitate standardized evaluation for future research?
>
> A: In our subsequent work, we will construct a comprehensive image dataset encompassing various perturbation techniques, including noise, blur, compression, color distortion, real-world degradation modeling, and diffusion model-based degradation. This dataset incorporates both individual and composite attack methodologies.
>
> We sincerely appreciate your comments and believe that our explanations and extra experiment results can effectively address your concerns. We would be grateful if you could consider raising the scores.
>
> [1] Qu, Chenfan, et al. "Towards modern image manipulation localization: A large-scale dataset and novel methods." *Proceedings of the IEEE/CVF Conference on Computer Vision and Pattern Recognition*. 2024.
>
> [2] Su, Lei, et al. "Can we get rid of handcrafted feature extractors? sparsevit: Nonsemantics-centered, parameter-efficient image manipulation localization through spare-coding transformer." *Proceedings of the AAAI Conference on Artificial Intelligence*. Vol. 39. No. 7. 2025.
>
> [3] Zhang, Quan, et al. "IMDPrompter: Adapting SAM to image manipulation detection by cross-view automated prompt learning." *arXiv preprint arXiv:2502.02454* (2025).
>
> [4] Xu, Zhipei, et al. "Fakeshield: Explainable image forgery detection and localization via multi-modal large language models." *arXiv preprint arXiv:2410.02761* (2024).

---

### Official Review · Reviewer_MEMz · 2025-10-30

**Soundness:** 2
**Presentation:** 2
**Contribution:** 1
**Rating:** 4
**Confidence:** 4

**Summary:**

The paper proposes a multi-scale fusion network for forgery detection. While the problem is relevant, I have several major concerns about the paper's technical novelty, the significance of its proposed contributions, and the thoroughness of its experimental evaluation. The core methodology seems to be an incremental combination of existing techniques, and the justification for certain design choices is not convincing. Furthermore, the evaluation is incomplete and lacks critical analysis. For these reasons, I cannot recommend acceptance of the paper in its current form.

**Strengths:**

The topic is meaningful. This paper is well-structured.

**Weaknesses:**

1. The core motivation and technical approach of the paper， fusing multi-scale features from RGB and noise streams， lacks novelty. This paradigm has been extensively explored in many prior works [r1-r2]. In the current state of research (as of 2024/2025), simply applying this fusion strategy is no longer considered a novel contribution. The paper fails to differentiate itself sufficiently from the large body of existing literature that employs similar principles.

[r1] Pixel-Inconsistency Modeling for Image Manipulation Localization. TPAMI 25

[r2] MVSS-Net: Multi-View Multi-Scale Supervised Networks for Image Manipulation Detection. TPAMI 22

2. The authors position the "Weight-aware Aggregation Decoder" (WAD) as a key contribution for adaptively weighting features from different scales. However, I am concerned about its actual technical significance. If my understanding is correct, the goal of WAD is to learn the importance of different feature streams. This same objective can be readily achieved using simpler and well-established methods. For instance, one could concatenate the features from all views along the channel dimension and then apply a standard 1x1 or 3x3 convolution, which would inherently learn to weigh and fuse the channels. Alternatively, a channel attention mechanism like SENet could be employed to explicitly model channel-wise interdependencies. The fact that these simpler alternatives exist casts doubt on the necessity and novelty of the proposed WAD module.

3. The experimental evaluation appears to focus exclusively on pixel-level localization metrics (e.g., pixel-wise F1-score or IoU). However, the paper fails to report image-level detection results (i.e., the binary classification performance of discriminating between real and manipulated images). This is a critical omission, as many real-world applications require a fast, image-level decision before performing expensive pixel-level analysis. A comprehensive evaluation must include standard image-level metrics like AUC or accuracy.

4. The proposed method involves multiple image resizing operations. It is well-documented that resizing algorithms (e.g., bilinear, bicubic interpolation) introduce their own high-frequency artifacts. These artifacts could act as confounding signals, potentially being learned by the detector instead of the actual manipulation traces. This raises two questions: (1) Have the authors analyzed or considered the negative impact of these resizing-induced artifacts on the detection performance? (2) Could the authors specify which interpolation method was used for resizing and provide a justification for this choice? Different methods produce different artifacts, which could affect reproducibility and performance.

5. The paper reports F1-scores as a primary metric, but it is not specified what decision threshold was used to compute them. Was a fixed threshold of 0.5 used for the probability maps, or was an optimal threshold selected for each model/dataset based on a validation set? This information is crucial for fair comparison and reproducibility.

6. The results show that adding a noise stream causes a significant performance degradation for the VLWA-Net baseline. The paper acknowledges this but dismisses it with a very brief explanation. Such a counterintuitive and dramatic result warrants a much more rigorous and in-depth analysis. Why does a model supposedly designed for this domain fail so catastrophically when provided with what should be additional, useful information? A deeper investigation into this phenomenon is necessary to build confidence in the authors' experimental methodology and conclusions.

**Questions:**

See Weakness.

---

> ### Author Response · Authors · 2025-11-21
> **Response to Reviewer MEMz**
>
> # Response to Reviewer MEMz
>
> Thank you for your comments. We will respond to your questions point by point.
>
> Q1: The core motivation and technical approach of the paper， "fusing multi-scale features from RGB and noise streams"， lacks novelty. This paradigm has been extensively explored in many prior works [r1-r2]. In the current state of research (as of 2024/2025), simply applying this fusion strategy is no longer considered a novel contribution. The paper fails to differentiate itself sufficiently from the large body of existing literature that employs similar principles.
>
> A: "Fusing multi-scale features from RGB and noise streams" is not our contribution. Extensive experiments and research have demonstrated that the dual-branch paradigm of "RGB+Noise" is highly suitable for image tampering localization tasks, which is why we have adopted it. Our innovations primarily focus on the overall framework, components, and training methods. The proposed "VAE+MDAM" combination excels at extracting more comprehensive and highly discriminative tampering artifacts compared to other models following the same paradigm. The introduced DFFM and WAD modules can effectively aggregate multi-scale features and decode the tampered regions from them. The training strategy with joint information supervision guides the model to learn complex tampering patterns from multiple dimensions. Experiments also confirm that our overall framework achieves state-of-the-art (SOTA) performance. These aspects constitute the key distinctions between our approach and the majority of existing models.
>
>
>
> Q2: The authors position the "Weight-aware Aggregation Decoder" (WAD) as a key contribution for adaptively weighting features from different scales. However, I am concerned about its actual technical significance. If my understanding is correct, the goal of WAD is to learn the importance of different feature streams. This same objective can be readily achieved using simpler and well-established methods. For instance, one could concatenate the features from all views along the channel dimension and then apply a standard 1x1 or 3x3 convolution, which would inherently learn to weigh and fuse the channels. Alternatively, a channel attention mechanism like SENet could be employed to explicitly model channel-wise interdependencies. The fact that these simpler alternatives exist casts doubt on the necessity and novelty of the proposed WAD module.
>
> A: WAD can dynamically adjust the decoding strategy based on the sensitivity of features at different scales and feature points within the same scale to the prediction results. a channel attention mechanism like SENet. In this scenario, the channel attention mechanism is confined to the sensitivity of features across different scales, and fails to capture the differences between spatial locations within features of the same scale. A simple convolution is ill-suited to cope with such complex and diverse tampering patterns. We conducted two additional ablation studies to compare the performance of the three decoders. The results (pixel-F1) demonstrate that our proposed WAD achieves the best performance.
>
> | Decoder    | CASIA1.0  | Columbia  | NIST16    | Coverage  | DSO       | Avg.F1    |
> | ---------- | --------- | --------- | --------- | --------- | --------- | --------- |
> | WAD (Ours) | **0.811** | 0.961     | **0.584** | **0.619** | **0.751** | **0.745** |
> | Conv       | 0.772     | 0.922     | 0.550     | 0.499     | 0.671     | 0.683     |
> | SE Block   | 0.772     | **0.963** | 0.542     | 0.546     | 0.729     | 0.710     |

---

> ### Author Response · Authors · 2025-11-21
> **Response to Reviewer MEMz**
>
> # Response to Reviewer MEMz
> Q3: The experimental evaluation appears to focus exclusively on pixel-level localization metrics (e.g., pixel-wise F1-score or IoU). However, the paper fails to report image-level detection results (i.e., the binary classification performance of discriminating between real and manipulated images). This is a critical omission, as many real-world applications require a fast, image-level decision before performing expensive pixel-level analysis. A comprehensive evaluation must include standard image-level metrics like AUC or accuracy.
>
> A: We added a detection head to our original model and retrained it using the MVSS protocol. We selected MVSS-Net, PSCC-Net, and TruFor as baselines (since only these models possess image-level classification capability) and retrained them on the same dataset. The models were evaluated on CASIA1.0, Columbia, Coverage, NIST16, and DSO as test sets. The first three contain both authentic and tampered images, while the last two contain only tampered images. The image-level F1-scores and Accuracy are reported in the two tables below. The results demonstrate that our model significantly outperforms these baseline models on both metrics.
>
> | Method (Image-F1) | CASIA1.0  | Columbia  | Coverage  | NIST16    | DSO       | Avg.F1    |
> | ----------------- | --------- | --------- | --------- | --------- | --------- | --------- |
> | MVSS-Net          | 0.697     | 0.636     | 0.667     | **1.000** | **1.000** | 0.800     |
> | PSCC-Net          | 0.581     | 0.709     | 0.641     | 0.971     | **1.000** | 0.780     |
> | TruFor            | 0.336     | 0.066     | 0.578     | 0.826     | 0.930     | 0.547     |
> | VLWA-Net          | **0.819** | **0.714** | **0.675** | 0.994     | **1.000** | **0.841** |
>
> | Method (Image-Acc) | CASIA1.0  | Columbia  | Coverage  | NIST16    | DSO       | Avg.Acc   |
> | ------------------ | --------- | --------- | --------- | --------- | --------- | --------- |
> | MVSS-Net           | 0.535     | 0.466     | 0.500     | **1.000** | **1.000** | 0.700     |
> | PSCC-Net           | 0.646     | 0.618     | 0.485     | 0.943     | **1.000** | 0.738     |
> | TruFor             | 0.224     | 0.181     | 0.425     | 0.704     | 0.870     | 0.481     |
> | VLWA-Net           | **0.797** | **0.644** | **0.590** | 0.988     | **1.000** | **0.804** |
>
> Q4: The proposed method involves multiple image resizing operations. It is well-documented that resizing algorithms (e.g., bilinear, bicubic interpolation) introduce their own high-frequency artifacts. These artifacts could act as confounding signals, potentially being learned by the detector instead of the actual manipulation traces. This raises two questions: (1) Have the authors analyzed or considered the negative impact of these resizing-induced artifacts on the detection performance? (2) Could the authors specify which interpolation method was used for resizing and provide a justification for this choice? Different methods produce different artifacts, which could affect reproducibility and performance.
>
> A: Nearly all IML models operate on a fixed input resolution, necessitating a resizing operation that adapts images of varying sizes. This process inevitably introduces some extraneous artifacts. However, since our model is designed to extract features sensitive to manipulation traces, it can, to some extent, mitigate the impact of these irrelevant artifacts. We employ bilinear interpolation to perform the resizing. This method is preferred because of its ability to achieve a superior balance between computational cost and output quality, effectively avoiding the coarse pixellation of the nearest-neighbor approach by generating smoother images with more natural edge transitions.
>
>
>
>
>
> Q5: The paper reports F1-scores as a primary metric, but it is not specified what decision threshold was used to compute them. Was a fixed threshold of 0.5 used for the probability maps, or was an optimal threshold selected for each model/dataset based on a validation set? This information is crucial for fair comparison and reproducibility.
>
> A: This was an oversight on our part. All our experiments were conducted using the standard threshold of 0.5. We have now addressed this point in the latest version of our manuscript.

---

> ### Author Response · Authors · 2025-11-22
> **Response to Reviewer MEMz**
>
> # Response to Reviewer MEMz
> Q6: The results show that adding a noise stream causes a significant performance degradation for the VLWA-Net baseline. The paper acknowledges this but dismisses it with a very brief explanation. Such a counterintuitive and dramatic result warrants a much more rigorous and in-depth analysis. Why does a model supposedly designed for this domain fail so catastrophically when provided with what should be additional, useful information? A deeper investigation into this phenomenon is necessary to build confidence in the authors' experimental methodology and conclusions.
>
> A: Perhaps we misunderstood your point. The paper **does not provide factual evidence corresponding to the so-called "The results show that adding a noise stream causes a significant performance degradation for the VLWA-Net baseline."** On the contrary, **adding a noise stream improves the performance of the VLWA-Net baseline.** For clarification, the noise branch was maintained throughout all ablation studies and was never added or removed. Our ablation experiments were conducted solely on the GAES, WAD, and JIS components. In Table 2, the first row represents our method (VLWA-Net), while the subsequent rows indicate the results after replacing the GAES, WAD, and JIS components, respectively. If the confusion persists, could you please indicate the specific part of the experimental data you are referring to? We would be happy to provide further clarification.
>
>
>
> We sincerely appreciate your comments and believe that our explanations and extra experiment results can effectively address your concerns. We would be grateful if you could consider raising the scores.

---

### Official Review · Reviewer_BKH1 · 2025-11-01

**Soundness:** 4
**Presentation:** 3
**Contribution:** 4
**Rating:** 8
**Confidence:** 5

**Summary:**

This paper tackles the persistent challenges in Image Manipulation Localization (IML). To address these issues, the authors introduce VLWA-Net, a robust framework leveraging VLMs through a fine-tuned VAE and a MDAM to capture the diverse and complex artifacts. The framework further employs a Weight-Aware Decoder (WAD) that adaptively weights features across scales and within scales to enhance localization precision. Extensive experiments on multiple benchmarks show that VLWA-Net outperforms state-of-the-art models in accuracy and robustness.

**Strengths:**

1. The article provides an in-depth analysis of the current challenges in IML tasks, such as the diversity of artifacts.
2. Proposed combination of VAE and MDAM effectively extracts diverse tampering artifacts.
3. Proposed WAD effectively decodes multi-scale features through adaptive weighting.
4. The framework is evaluated across seven benchmark datasets, and robustness is tested under common distortions (JPEG, blur, noise).
5. The idea is straightforward and easy to implement.

**Weaknesses:**

1. Some technical details are not clearly explained, particularly regarding the MDAM and WAD modules.
2. The backbone among all the methods listed in Table.1 are different, making the comparison unfair.
3. Fail to compare with more effective models such as APSC-Net, or discuss them in related works, limiting the demonstrated effectiveness.
4. Could the authors provide the actual dimensionalities of the feature maps from GAES and NTS at each scale, and explain how alignment is ensured before fusion in DFFM? This is important because scale alignment across architectures with different receptive fields (e.g., ConvNeXt vs. ViT) can significantly impact fusion effectiveness.

[1] Qu C, Zhong Y, Liu C, et al. Towards modern image manipulation localization: A large-scale dataset and novel methods[C]//Proceedings of the IEEE/CVF Conference on Computer Vision and Pattern Recognition. 2024: 10781-10790.

**Questions:**

Please provide a detailed explanation of the working mechanism of the SCFF module.

---

> ### Author Response · Authors · 2025-11-21
> **Response to Reviewer BKH1**
>
> # Response to Reviewer BKH1
>
> Q1: Some technical details are not clearly explained, particularly regarding the MDAM and WAD modules.
>
> A: Figure 2 presents the detailed architecture of MDAM. In Section 3.2, we provide a comprehensive explanation of its specific structure, working mechanism, and functionalities—including dilated convolution, DCT transformation, and the dual spatial-frequency branch design. The MDAM is to enrich the scale diversity of the features extracted by the VAE and enhance both spatial and frequency components to capture more tampering traces.
>
> Figure 3 illustrates the internal architecture of WAD. In Section 3.5, we provide a detailed exposition of WAD's motivation, structure, working principles, and functionalities. WAD introduces four groups of learnable 3D weight parameters that can adaptively capture the sensitivity differences between features across different scales, as well as among pixels within the same scale.
>
>
>
> Q2: The backbone among all the methods listed in Table.1 are different, making the comparison unfair.
>
> A: The comparison methods in Table 1 employ distinct backbones. For instance, MVSS-Net and CAT-Net utilize CNNs as their backbone, TruFor adopts a ViT backbone, while Mesorch features a hybrid architecture combining both. However, the SOTA performance comparison experiments evaluate the overall performance across different models rather than analyzing the impact of the individual backbones on system performance. The results in Table 1 demonstrate that our framework achieves significantly superior overall performance compared to other models. In the ablation study, we have investigated the impact of backbone selection on the performance of the overall  framework. We replace the VLMs-based backbone in VLWA-Net with alternative backbones based on CNNs and Transformers, followed by performance evaluations. The results, as shown in settings 1, 2, and 3 in Table 2, indicate that our VLMs-based backbone achieves an improvement of approximately 83.5% over ConvNeXt and about 26.9% over SegFormer. This observation underscores that employing a VLM-based backbone contributes to substantial overall performance gains, while the choice between CNN- or ViT-based backbones exhibits relatively minor impact on the final performance.
>
> Q3: Fail to compare with more effective models such as APSC-Net, or discuss them in related works, limiting the demonstrated effectiveness.
>
> A: We retrained APSC-Net[1] and SparseViT[2] using the same training settings, and the test results are as follows (pixel-F1). The results for IMDPrompter[3] and FakeShield[4] are cited from the original paper.
>
> | Method         | CASIA1.0  | Columbia  | NIST16    | Coverage  | DSO       | Avg.F1    |
> | -------------- | --------- | --------- | --------- | --------- | --------- | --------- |
> | APSC-Net       | 0.798     | 0.941     | 0.500     | 0.402     | 0.617     | 0.652     |
> | SparseViT      | **0.827** | 0.959     | 0.384     | 0.513     | 0.239     | 0.584     |
> | IMDPrompter    | 0.763     | 0.873     | 0.411     | **0.636** | \         | 0.671     |
> | FakeShield     | 0.600     | 0.750     | 0.370     | \         | 0.52      | 0.560     |
> | VLWA-Net(Ours) | 0.811     | **0.961** | **0.584** | 0.619     | **0.751** | **0.745** |
>
> The results indicate that our method significantly outperforms these models in terms of average performance.

---

> > ### Comment · Reviewer_BKH1 · 2025-11-23
> > **Response to Authors Rebuttal**
> >
> > Thank you for your response, please add the new experiment results to table1, add submit the updated paper before the end of the rebuttal stage.

---

> ### Author Response · Authors · 2025-11-21
> **Response to Reviewer BKH1**
>
> Q4: Could the authors provide the actual dimensionalities of the feature maps from GAES and NTS at each scale, and explain how alignment is ensured before fusion in DFFM? This is important because scale alignment across architectures with different receptive fields (e.g., ConvNeXt vs. ViT) can significantly impact fusion effectiveness.
>
> A: In our paper, we specify the feature dimensions at several critical data nodes using mathematical notation. Specifically, the output dimension of the VAE is fixed at 256×64×64. Both GAES and NTS generate tampering features at four scales, with identical dimensions for their outputs. For a 512×512 input image, the output feature dimensions of both GAES and NTS are as follows:128×128×128、256×64×64、512×32×32 and 1024×16×16. Regardless of the input image resolution, the number of feature channels remains fixed.
>
> The MDAM is responsible for performing both scale-wise and feature-space alignment on the features output by the VAE. Specifically, the adaptive feature pyramid within MDAM transforms the single-scale input features into multi-scale representations, ensuring dimensional consistency with the features extracted by ConvNeXt. In each frequency-spatial enhancement block, we employ a dilated convolution and a fusion convolution to reconstruct and refine features across different scales. This architecture effectively constructs a unified multi-scale feature space that bridges ViT and ConvNeXt representations.
>
> Q5: Please provide a detailed explanation of the working mechanism of the SCFF module.
>
> A: We provide a detailed description of how DFFM and SCFF work in Appendix.3.
>
> We sincerely appreciate your comments and believe that our explanations and extra experiment results can effectively address your concerns. We would be grateful if you could consider raising the scores.
>
> [1] Qu, Chenfan, et al. "Towards modern image manipulation localization: A large-scale dataset and novel methods." *Proceedings of the IEEE/CVF Conference on Computer Vision and Pattern Recognition*. 2024.
>
> [2] Su, Lei, et al. "Can we get rid of handcrafted feature extractors? sparsevit: Nonsemantics-centered, parameter-efficient image manipulation localization through spare-coding transformer." *Proceedings of the AAAI Conference on Artificial Intelligence*. Vol. 39. No. 7. 2025.
>
> [3] Zhang, Quan, et al. "IMDPrompter: Adapting SAM to image manipulation detection by cross-view automated prompt learning." *arXiv preprint arXiv:2502.02454* (2025).
>
> [4] Xu, Zhipei, et al. "Fakeshield: Explainable image forgery detection and localization via multi-modal large language models." *arXiv preprint arXiv:2410.02761* (2024).

---

### Official Review · Reviewer_kroU · 2025-11-01

**Soundness:** 1
**Presentation:** 3
**Contribution:** 1
**Rating:** 2
**Confidence:** 4

**Summary:**

The paper proposes VLWA-Net, a framework for image manipulation localization that allegedly leverages Vision-Language Models (VLMs) to extract tampering artifacts. The model integrates a “VLMs-enhanced Artifact Extractor,” a Multi-Domain Artifact Modulator (MDAM), and a Weight-Aware Decoder (WAD). The authors claim that by incorporating VLMs, the method achieves superior robustness and generalization across multiple benchmarks.

**Strengths:**

- The overall architecture is clearly described and experimentally validated.
- Experiments are comprehensive across several datasets.

**Weaknesses:**

## Main Issue
- The central concept is flawed: the paper repeatedly calls SAM a Vision-Language Model, but SAM is a vision-only segmentation model, not trained with textual input or multimodal supervision.
- Claims such as “VLMs exhibit superior universal feature extraction capabilities” (lines 50–51) are unsupported by evidence.
- The proposed “VLMs-enhanced Artifact Extractor” could be replaced by any visual backbone (e.g., DINOv3, ViT) without affecting the framework.
- The work’s framing around VLMs is misleading and self-inconsistent, as “language” plays no role in the model.

## Minor Issue
- Using VAE as a module name is highly inappropriate, as it conflicts with the well-established term Variational Autoencoder in ML literature.


In summary, I believe the core argument of this paper is invalid, and the discussions built around it fail to support the claimed contributions to the network and the task. The paper lacks solid theoretical and logical foundations; therefore, I **strongly recommend rejection.**

**Questions:**

As discussed in the Weaknesses section, the paper’s core argument is not well-founded.

---

> ### Author Response · Authors · 2025-11-21
> **Response to Reviewer kroU**
>
> # Response to Reviewer kroU
>
> Thank you for your comments. We will respond to your questions point by point.
>
> Q1: The central concept is flawed: the paper repeatedly calls SAM a Vision-Language Model, but SAM is a vision-only segmentation model, not trained with textual input or multimodal supervision.
>
> A: In the second section of the original SAM paper[1], it mentions that ". . . where a prompt can be a set of foreground/background points, a rough box or mask, free-form text . . ." and Figure 4 explicitly indicates that the "prompt encoder" accepts three different types of prompts: points, boxes, and text. This demonstrates that SAM can accept both images and text as input, and therefore, SAM can be regarded as a type of Vision-Language Model (VLM), even though its output consists solely of masks.
>
>
>
> Q2: Claims such as “VLMs exhibit superior universal feature extraction capabilities” (lines 50–51) are unsupported by evidence.
>
> A: Indeed, we did not provide substantial evidence for this in the original paper. We have already supplemented this in our latest paper. Papers [1], [2], [3], and [4] all mention that VLMs possess powerful feature extraction capabilities. Furthermore, the image encoders within VLMs are generally composed of numerous stacked Vision Transformer (ViT) blocks, resulting in a massive number of parameters, and they are trained on extremely large-scale image datasets. Consequently, these image encoders can be adapted to various downstream tasks. In our ablation studies, replacing the VAE with other non-VLM visual backbones led to a significant performance drop. This further proves that the features extracted by VLMs are more discriminative and better at capturing diverse tampering artifacts.
>
>
>
>
>
> Q3: The proposed “VLMs-enhanced Artifact Extractor” could be replaced by any visual backbone (e.g., DINOv3, ViT) without affecting the framework.
>
> A: The replaceability of the VAE with other visual backbones demonstrates the flexibility of our framework. However, the performance decline observed when using backbones like ConvNeXt or SegFormer highlights our success in fully leveraging the capabilities of VLMs for the image manipulation localization task.
>
> The ablation experimental results related to the backbone are as follows (pixel-F1):
>
> | Number | Backbone            | Avg.F1 |
> | ------ | ------------------- | ------ |
> | 1      | SAM-ViT-L(Ours)     | 0.745  |
> | 2      | SAM-ViT-B           | 0.702  |
> | 3      | CLIP-ViT-L/14       | 0.550  |
> | 4      | CLIP-ViT-L/14@336px | 0.559  |
> | 5      | Dinov3              | 0.654  |
> | 6      | ConvNeXt            | 0.406  |
> | 7      | SegFormer           | 0.587  |
>
> These results demonstrate that SAM is the most suitable backbone for our framework, and our design successfully unleashes SAM's potential for the IML task.
>
> Q4: The work’s framing around VLMs is misleading and self-inconsistent, as “language” plays no role in the model.
>
> A: It is true that we did not utilize the text modality. Our work is primarily centered on harnessing the visual capabilities of VLMs, which we believe rightfully qualifies it under the "VLMs" scope as indicated in the paper. Furthermore, the related work section also emphasizes the visual-task competencies of VLMs.
>
>
>
> Q5: Using VAE as a module name is highly inappropriate, as it conflicts with the well-established term Variational Autoencoder in ML literature.
>
> A: We will revise the abbreviation "VAE" (e.g., VeAE) to avoid any potential confusion.
>
> We sincerely appreciate your comments and believe that our explanations and extra experiment results can effectively address your concerns. We would be grateful if you could consider raising the scores.
>
> [1] Kirillov, Alexander, et al. "Segment anything." *Proceedings of the IEEE/CVF international conference on computer vision*. 2023.
>
> [2] Zhang, Quan, et al. "IMDPrompter: Adapting SAM to image manipulation detection by cross-view automated prompt learning." *arXiv preprint arXiv:2502.02454* (2025).
>
> [3]Li, Zongxia, et al. "A survey of state of the art large vision language models: Alignment, benchmark, evaluations and challenges." *arXiv preprint arXiv:2501.02189* (2025).
>
> [4]Zhang, Haifeng, et al. "Towards Universal AI-Generated Image Detection by Variational Information Bottleneck Network." *Proceedings of the Computer Vision and Pattern Recognition Conference*. 2025.
>
> [5]Sun, Ke, et al. "Towards general visual-linguistic face forgery detection." *Proceedings of the Computer Vision and Pattern Recognition Conference*. 2025.

---

> > ### Comment · Reviewer_kroU · 2025-11-28
> > **A pure visual setting paper but claim as its contribution from Vision-language model**
> >
> > Thanks for the authors' response. In response to the authors’ reply, I would like to clarify that my concerns in Q1 and Q3 are essentially connected: **the paper does not utilize the text modality at all**, meaning that the backbone could be replaced by any visual model without changing the nature of the method. Therefore, the claimed advantages of “VLMs” are irrelevant to the actual contribution, since the task and the proposed framework operate entirely within **a purely visual setting**.
> >
> > Moreover, although SAM can technically accept limited text-like prompts and consider the claim of "SAM is a VLM is correct", the authors’ own experiments in rebuttal show that other genuine VLMs such as CLIP perform worse than purely visual backbones like DINO and SegFormer. This indicates that the performance gain comes from SAM itself, not from VLMs as a general category, and thus weakens the central narrative built around VLM superiority.
> >
> > In addition, the motivation and fusion design remain vague and largely intuitive. The module connections—mainly fusion between some neural networks—lack clear technical justification, formal analysis, or rigorous comparative evidence supporting why these choices are necessary or effective. Consistent with Reviewer MEMz’s earlier comments, the proposed fusion strategy does not establish a meaningful technical gap from prior works such as PIM or MVSS-Net.
> >
> > For these reasons, I still believe the paper’s core argument is difficult to sustain, its contribution to the community is limited, and its framing may even mislead future researchers. Overall, the work does not yet meet the standards for acceptance at ICLR.

---

> > > ### Author Response · Authors · 2025-11-30
> > > **Response to Reviewer kroU**
> > >
> > > # Response to Reviewer kroU
> > > Regarding your response, I must clarify the following points:
> > >
> > > - Our method focuses on the **visual capabilities of VLMs**, aiming to leverage their powerful and **general visual feature extraction abilities** to capture comprehensive and highly discriminative tampering features. This approach does not utilize the text modality, fully aligning with the meaning of the term "VLMs".
> > >
> > > - When using CLIP as the backbone, the model's relatively lower performance is primarily attributed to its fixed input resolution of 224×224, whereas other backbones (e.g., Mesorch, MVSS-Net) support higher input resolutions such as 512×512 or 1024×1024. Based on our extensive experiments in image manipulation localization (IML), we have consistently observed that reducing input resolution leads to significant performance degradation. This is because lower resolution results in substantial loss of image details, making tampering artifacts more subtle and difficult to detect. Additionally, the pre-training task of CLIP makes it more suitable for classification tasks rather than segmentation tasks. Despite these limitations, CLIP still achieves an average F1-score of 0.55, which sufficiently demonstrates the effectiveness of our proposed method. Additionally, as shown in Table 1 of our paper, **using only SAM achieves an average F1-score of only 0.575**, which is not a strong performance. In contrast, our VLWA-Net attains an average F1-score of 0.745, significantly outperforming the SAM-only approach. This clearly demonstrates that our framework's effectiveness does not rely solely on SAM, but rather stems from the synergistic contribution of all its components.
> > >
> > > - Our paper and the rebuttal to the reviewers include extensive ablation studies and visual analyses (provided in the appendix) on **all the proposed components**. These results sufficiently demonstrate the necessity and effectiveness of our design choices.

---

### Note · Authors · 2026-05-07

I have read and agree with the venue's withdrawal policy on behalf of myself and my co-authors.

---

### Meta-Review · Area_Chair_9qq1 · 2026-01-02

**Summary:**

This submission proposes VLWA-Net, a framework for image manipulation localization that leverages Vision-Language Models  with a VLMs-enhanced Artifact Extractor, Multi-Domain Artifact Modulator, and Weight-Aware Decoder.

The reviewers have raised several significant concerns:

Reviewer kroU identifies a fundamental flaw in the paper's core concept - the characterization of SAM as a Vision-Language Model when it operates in a purely visual setting without leveraging language capabilities.

Reviewer MEMz questions the technical novelty, noting that the fusion paradigm lacks innovation and that the proposed WAD module could be replaced by simpler alternatives.

Reviewer BKH1 provides a positive assessment but acknowledges comparison fairness issues.

Reviewer BeuY raises concerns about limited evaluation scope and shallow theoretical justification.

**Reviewer Concerns:**

The authors' rebuttal addressed several technical points raised by reviewers, particularly providing additional experiments and clarifications about threshold selection, image-level evaluation, and comparisons with additional models.

However, the most critical concerns remain unresolved. Reviewer kroU's fundamental objection about the misleading framing around VLMs persists, i.e., the paper does not utilize language capabilities, and the performance gains appear to stem from SAM's visual features rather than any VLM-specific advantages. The rebuttal's claim that SAM "can be regarded as a type of VLM" because it can technically accept text prompts does not address the core issue that the method operates entirely in a visual modality.

Reviewer MEMz's concerns about novelty and technical significance of the WAD module also remain outstanding, as the authors' response comparing WAD to simpler alternatives shows only incremental improvements rather than demonstrating fundamental innovation. The lack of clear differentiation from existing fusion-based approaches undermines the paper's contribution claim.

**Reviewer Scores:**

Based on the discussion and rebuttal quality, I anticipate the following score adjustments if reviewers had participated fully:

Reviewer kroU would maintain their score of 2, as the core conceptual issue remains unresolved.

Reviewer MEMz might slightly increase their score from 4 to 5, acknowledging the additional experiments but maintaining concerns about novelty.

Reviewer BKH1 would likely maintain their score of 8, as they were generally satisfied with the response.

Reviewer BeuY would likely maintain their score of 6, as their concerns about limited evaluation scope and theoretical depth were not fully addressed.

The average remains below the acceptance threshold, with two reviewers expressing fundamental concerns about the paper's core contributions and framing.

---

### Decision · Program_Chairs · 2026-01-26

Reject